# Textile Antenna Sensor in SIW Technology for Liquid Characterization

**DOI:** 10.3390/s23187835

**Published:** 2023-09-12

**Authors:** Mariam El Gharbi, Maurizio Bozzi, Raúl Fernández-García, Ignacio Gil

**Affiliations:** 1Department of Electronic Engineering, Universitat Politècnica de Catalunya, 08222 Terrassa, Spain; raul.fernandez-garcia@upc.edu; 2Department of Electrical, Computer and Biomedical Engineering, University of Pavia, 27100 Pavia, Italy; maurizio.bozzi@unipv.it

**Keywords:** liquid characterization, textile antenna sensor, substrate-integrated waveguide (SIW), dielectric properties, quality factor, circular cavity

## Abstract

This study showcases the creation of an innovative textile antenna sensor that utilizes a resonant cavity for the purpose of liquid characterization. The cavity is based on circular substrate integrated waveguide (SIW) technology. A hole is created in the middle of the structure where a pipe is used to inject the liquid under test. The pipe is covered by a metal sheath to enhance the electromagnetic field’s penetration of the tube, thus increasing the device’s sensitivity. The resonance frequency of the proposed system is altered when the liquid under test is inserted into the sensitive area of the structure. The sensing of the liquid is achieved by the measurement of its dielectric properties via the perturbation of the electric fields in the SIW configuration. The S_11_ measurement enables the extraction of the electromagnetic properties of the liquid injected into the pipe. Specifically, the dielectric constant of the liquid is determined by observing the resonance frequency shift relative to that of an air-filled pipe. The loss tangent of the liquid is extracted by comparing the variation in the quality factor with that of an air-filled pipe after eliminating the inherent losses of the structure. The proposed SIW antenna sensor demonstrates a high sensitivity of 0.7 GHz/Δ*ε_r_* corresponding to a dielectric constant range from 4 to 72. To the best of our knowledge, this article presents for the first time the ability of a fully textile SIW cavity antenna-based sensor to characterize the dielectric properties of a liquid under test and emphasizes its differentiating features compared to PCB-based designs. The unique attributes of the textile-based antenna stem from its flexibility, conformability, and compatibility with various liquids.

## 1. Introduction

In recent years, the demand for novel and low-cost sensors for the characterization of liquids has created significant interest in research due to their wide application in various industries including environmental monitoring, pharmaceuticals, and food and beverage [1]. The capability to properly measure and detect liquid properties, such as concentration, is critical to ensure process control, product quality, and environmental safety. There exist different types of measurement techniques for liquid characterization, including transmission-line techniques [2], coaxial probes [3], free space methods [4], and cavity resonator methods [5]. Among these methods, the cavity resonators are well-suited for precise and narrowband characterization, offering a high level of accuracy [6]. Cavity resonators find great benefits in substrate-integrated waveguide (SIW) technology, offering cost-effective solutions and a relatively high-quality factor. SIW technology is a popular technique for designing microwave systems due to its advantages in cost-effectiveness, miniaturization, and integration capabilities [7,8].

Indeed, substrate-integrated waveguide (SIW) technology has made significant advancements in recent years. Several sensors based on SIW cavities have been proposed [9,10,11]. These resonators are typically designed on rigid planar substrates, enabling easy integration with other electronic components and systems. Some research has been reported in the literature, where SIW resonators were used to investigate the dielectric properties of liquids under test. For instance, a high-sensitivity sensor boasting micro-volume measurement is presented in [12], but it provided limited specificity and applicability for such a sensor. The authors in [13] demonstrated accurate measurements with a simple and low-cost system, but a large volume of the sample is required for the immersion technique for measuring. The aforementioned SIW cavity sensors are commonly implemented on rigid substrates, which are typically used in electronic systems. However, these traditional substrates are not suitable for electronic-textile (e-textile) applications.

The development of textile-based antennas for liquid characterization is entrenched in their exceptional flexibility and conformability. Unlike traditional PCB-based antennas, which are inherently rigid and often ill-suited for conforming to irregular shapes or directly interfacing with liquids, textile-based antennas offer remarkable pliability. This enables seamless integration onto various substrates, including curved surfaces and complex geometries, facilitating direct and efficient interaction with the liquids under scrutiny. In the present era, textiles are becoming the first choice for developing and fabricating wearable antennas and sensors due to their ability to offer the wearer comfort and flexibility [14,15]. In [16], a textile microwave resonator sounds intriguing and highly relevant. The integration of resonator technology into textile material presents a responsive system capable of sensing liquid. This could have a significant impact on various applications, including environmental monitoring, safety, and healthcare. Therefore, the convergence of textile and SIW technology has resulted in a remarkable fusion of wireless communications and liquid-sensing capabilities. SIW, an innovative transmission-line technology, enables the design and integration of various electromagnetic structures into low-cost substrates. The textile SIW antenna sensors merge the advantages of textile materials with the capabilities of SIW technology, resulting in lightweight, robust, and high-performance sensing solutions [17]. By integrating SIW technology with textile antenna-based sensors, we can now create smart fabrics that are able to distinguish and monitor liquids in real time. The textile antenna sensor represents a significant advancement in healthcare applications [18]. It combines a textile antenna-based sensor with SIW technology, allowing for simultaneous signal transmission and measurement of liquid properties. The principle behind the antenna-based sensor behavior lies in the interaction between the liquid under test (LUT) and electromagnetic waves. When LUT is injected into the sensing area of the SIW structure, it causes an electromagnetic field disturbance, which changes the antenna performance in terms of operation frequency and efficiency. This novel approach provides a non-destructive and versatile solution for liquid characterization, overcoming many limitations of conventional techniques.

This research aims to explore the design, fabrication, and characterization of a textile SIW antenna-based sensor for liquid analysis. The novelty of the proposed structure consists of the use of a full textile substrate applying the embroidery technique to implement a cost-effective SIW cavity resonator using conductive yarn. Furthermore, the whole structure of the SIW antenna-based sensor consists of a dielectric substrate (felt fabric) sandwiched between two conductive layers, which can be achieved by using Less EMF ShieldIt Super Fabric. The proposed structure is based on a SIW cavity antenna-based sensor with a hole located at its center and a pipe passing through, where the propagated EM field is concentrated, in order to achieve better sensing. In order to improve the extent to which the electromagnetic field enters the pipe, a metal sheath is utilized to encase the walls of the pipe. This, in turn, increases the sensitivity of the proposed system. This work uses an approach to retrieve the dielectric constant and the loss tangent of the LUT; the dielectric constant is obtained from the resonance frequency shift, whereas the loss tangent is derived from the quality factor after removing the actual losses of the structure. The remainder of the paper is organized as follows. Section 2 details the methodology of the proposed design, the sensing principle, and the materials used in the manufacturing. Section 3 provides the results of the proposed SIW antenna sensor. A comparison of the proposed SIW antenna sensor with previously reported works in the literature for liquid characterization is presented in Section 4. Finally, Section 5 exhibits the main conclusions.

## 2. Materials and Methods

The SIW structure proposed in this work consists of a circular resonant cavity with a hole in the middle containing a polypropylene (PP) pipe where the liquid under test can be injected. The proposed structure was designed by choosing appropriately spaced via holes with the same diameter. There are two design rules related to the via diameter (*D*) and the spacing between the vias (*S*), as given by [19]: $D<\lambda_{g}/5$ and S≤2D. The optimal space was determined using these rules to minimize the leakage loss of the SIW structure. The textile antenna-based sensor is composed of a top patch with a hole and a bottom ground. These two conductors are connected by vias to create the SIW structure. The conductors of the top and bottom planes are made of a nickel-plated Less EMF ShieldIt Super Fabric attached to the substrate using an iron. The conductive fabric has a thickness of 0.17 mm and a sheet resistance of 0.07 ohm/square and the vias are implemented using commercial Shieldex 117/17 dtex 2-ply conductor yarn. This yarn is made from 99% pure silver-plated nylon yarn 140/17 dtex with a linear resistance <30 Ω/cm. During the embroidery, the conductive yarn is carefully positioned on the substrate to define the shape and path of the desired vias. This can be achieved by using precision placement tools on the embroidery machine to ensure precise alignment. The proposed system is designed and simulated using CST Microwave Studio 2022 software at a 5.8 GHz operation frequency. The embroidery technique was used to manufacture the vias of the SIW structure. This method needs to export the simulated structure from CST software to a file format readable by the design embroidery software (Easy Design EX4.0). The computerized embroidery process was implemented using a satin fill pattern creating vias of stitches. The stitch uses a needle yarn that passes the substrate and overlaps with a bobbin yarn. The computerized embroidery machine ensures precise design replication of our SIW structure. For the SIW antenna sensor manufacturing, the embroidery machine Singer Futura XL 550 was used. The proposed structure is intended to be flexible, low profile, and light weight. In this work, the felt fabric is selected as a low-cost textile substrate for the fabrication of the SIW structure. A Split Post-Dielectric Resonator (SPDR) is used to characterize the dielectric properties of the felt substrate. The dielectric constant and loss tangent for the felt substrate are $\varepsilon_{r}=1.2$ and tan δ=0.0013, respectively. The geometrical dimensions of the proposed system are clearly illustrated in Figure 1. Table 1 presents the optimized geometrical parameters of the proposed structure.

A parametric study is performed using a geometric tuning approach on different parameters of the proposed antenna sensor in order to obtain better insight into the antenna’s physical performance and to fine-tune its behavior specifically for the 5.8 GHz frequency. Each S_11_ for the variation in the radius of the circular SIW (R) and the length of the antenna ($l_{a}$) is shown in Figure 2. The cavity resonates on the fundamental mode $TM_{01}$, which has a maximum electric field amplitude at its center. The proposed system is based on the principle of electromagnetic wave propagation and interaction with the liquid medium. When electromagnetic waves encounter a liquid, the dielectric properties of the liquid under test inside the cavity are modified. Consequently, this leads to a modification in the resonance frequency and the quality factor of the cavity. The scope of these interactions provides valuable information about the properties of the LUT. The center of the proposed sensor contains the PP pipe, where the electromagnetic field is highly concentrated at the edge of the hole, as illustrated in Figure 3a. Note that the pipe material is taken into consideration when designing and implementing the system to ensure it meets the exact requirements used in the measurement. To enhance the penetration of the EM field into the pipe, a metal sheath is employed to encase the pipe walls. This modification increases the sensitivity of the system (Figure 3b). This approach significantly improves the sensitivity of the proposed device to retrieve the dielectric properties of the LUT. During the experiment, a commercial aluminum conductive tape is used as a metal sheath to cover the PP pipe to ensure electrical contact. A photograph of the fabricated prototype of the proposed SIW antenna sensor is presented in Figure 4.

## 3. Results

The S_11_ of the SIW antenna sensor is measured when filling the pipe with different liquids. It is measured using a N9916A FieldFox microwave analyzer. Note that the antenna in our system has a gain of 3.25 dBi, which can be used for wireless communication. Figure 5 shows the measured S_11_ for four different cases (air, distilled water, pure isopropanol, and a mixture of 90% isopropanol and 10% water). From the obtained results, the resonance frequency of the SIW sensor decreases when different liquids are injected into the pipe, i.e., when increasing the dielectric constant of the liquid under test. To be precise, the resonance frequencies shift from 5.8 GHz for the air-pipe to 5.32 GHz for the pipe filled with water. Apart from the change in frequency shift, altering the liquid inside the pipe also impacts the quality factor of the SIW sensor. The resonance frequency shift can be used to retrieve the dielectric constant of the liquid under test. To attain this goal, CST simulations are executed, taking into account a liquid-filled tube exhibiting various dielectric constants ranging from 10 to 80. The aim is to compute the associated frequency shift denoted as “$\Delta_{f}$”. The outcomes of these evaluations are showcased in Figure 6. The simulation results were fitted using the curve-fitting technique, relating the frequency shift $\Delta_{f}$ to the dielectric constant $\varepsilon_{r}$ of the liquid under test, whose expression is:(1)$$\varepsilon_{r}=210.6\;{\left(\Delta_{f}\right)}^{2}+43.214\;\Delta_{f}+1.4536$$

In this way, this expression or the plot in Figure 5 enables the extraction of the dielectric constant of the liquid under test based on the measured frequency shift. The selection of this specific fitting function was based on its ability to achieve a good correlation $(R^{2}=0.9975).$ The example of the mixture of 10% Iso 90% Water is presented in Figure 5. Once the mixture is introduced into the pipe, the recorded change in frequency (compared to the scenario where the pipe is empty) amounts to a value of $\Delta_{f}$ = 0.44 GHz. According to the plot in Figure 6, the corresponding dielectric constant of the mixture is 61.24 (indicated in the plot). This process is employed for all liquids under test examined within this article. For comparative purposes, the dielectric properties of different liquids are measured using a commercial open-ended coaxial probe from a Keysight N1501A Dielectric Probe Kit, which is used as the reference technique.

The dielectric constant and the loss tangent of all the liquids under test are compared with the values of the dielectric properties measured by the coaxial probe. Table 2 displays a comparison between the dielectric constant values acquired through the presented method and those obtained using the coaxial probe characterization technique. The dielectric constant values listed in Table 2 are derived by factoring in loss effects, utilizing the approach outlined in [6]. The relative error is calculated for all the cases, as presented in Table 2, to provide the accuracy and precision of our system. The error in the determination of the dielectric constant is less than or at least equal to 19%. The average error of the proposed sensor is 8.25% for the dielectric constant retrieved. The error factor can arise from different aspects, such as discrepancies in geometry between the simulation models and the test model, or potential variations in the environment.

To obtain the loss tangent of the liquid being tested, we utilize changes in the unloaded quality factor. The procedure for deriving the loss tangent is detailed in reference [6]. Simulated values showcasing the relationship between the loss tangent and the unloaded quality factor are illustrated in Figure 7. Through the computation of the unloaded quality factor derived from the measured S_11_ for the liquid under test (LUT), it becomes possible to extract the loss tangent of the tested liquid. This can be accomplished using Figure 7, or by utilizing the equation acquired from the fitting technique.
(2)$$tan\;\delta =51.9927{\left(Q_{u}^{Sim}\right)}^{\left(-1.8089\right)}$$

This expression or the plot in Figure 7 enables the extraction of the loss tangent of the LUT based on the measured unloaded quality factor. For example, for the mixture of 10% Iso and 90% Water, the measured unloaded quality factor is 14.9. Thus, the corresponding loss tangent is 0.39 (indicated in the plot). The acquired value closely approximates the reference value obtained using the coaxial probe, which is 0.40. Table 3 represents the values of the loss tangent obtained from the presented method compared with the reference values measured by the coaxial probe characterization technique. For each mixture, the relative error is calculated, as presented in Table 3. The error in determining the loss tangent is either less than or equal to 12%. The average error of the retrieved loss tangent is 5.29%; this value provides an indication of the sensor’s overall performance in terms of measurement accuracy.

## 4. Discussion

Table 4 presents a comparison between the proposed system and previous works described in the literature for liquid characterization. The proposed sensor offers a distinct benefit compared to other systems. It introduces a groundbreaking feature by enabling the characterization of liquids using a fully textile SIW system. Note that the antenna used in the system is a broadband antenna that can be used to transmit data for the retrieved dielectric properties of the liquid under test. This part will be included in future work. The sensor sensitivity (*S*) is defined by *S* = $\Delta_{f}/\Delta \varepsilon_{r}$, where $\Delta_{f}$ is the shift in resonance frequency and $\Delta \varepsilon_{r}$ is the maximum detectable range of permittivity variation [20]. The proposed system has comparable performance to that of previous devices. However, it is important to highlight that the other systems are fabricated on rigid substrates, whereas the proposed SIW antenna sensor was entirely fabricated on textile using the embroidery technique. Using textiles as a substrate makes our system lightweight and portable, which is suitable for liquid characterization applications. It can be easily folded and rolled up without adding significant bulk or weight. This portability is useful in different areas, such as blood analysis. According to Table 4, the proposed SIW antenna sensor exhibits higher sensitivity than the reported microwave sensors. Numerous sensor types are proposed in the literature for liquid characterization, as indicated in Table 4. These sensors operate at different frequencies, and most of the works are implemented by printed circuit board (PCB). In this work, the novel manufacturing approach offers several advantages compared to traditional manufacturing technologies. These advantages include a low profile, flexibility, and light weight.

## 5. Conclusions

A textile SIW antenna sensor is implemented to characterize the dielectric properties of a liquid under test (LUT). The proposed system is designed, simulated, fabricated, and measured at a resonance frequency of 5.8 GHz. A pipe is introduced at the center of the SIW structure. This pipe serves as a conduit for injecting and extracting the LUT, with the primary goal of characterizing its electromagnetic properties. The proposed SIW sensor antenna is used to ascertain the dielectric constant and loss tangent of various liquid mixtures. The acquired values are compared with measurements taken using a coaxial probe. The proposed SIW antenna sensor presents a high sensitivity of 0.7 GHz/$\Delta \varepsilon_{r}$ corresponding to a dielectric constant range from 4 to 72. The average error of the retrieved values of the dielectric constant and loss tangent is 8.25% and 5.29%, respectively.

## Figures and Tables

**Figure 1 sensors-23-07835-f001:**
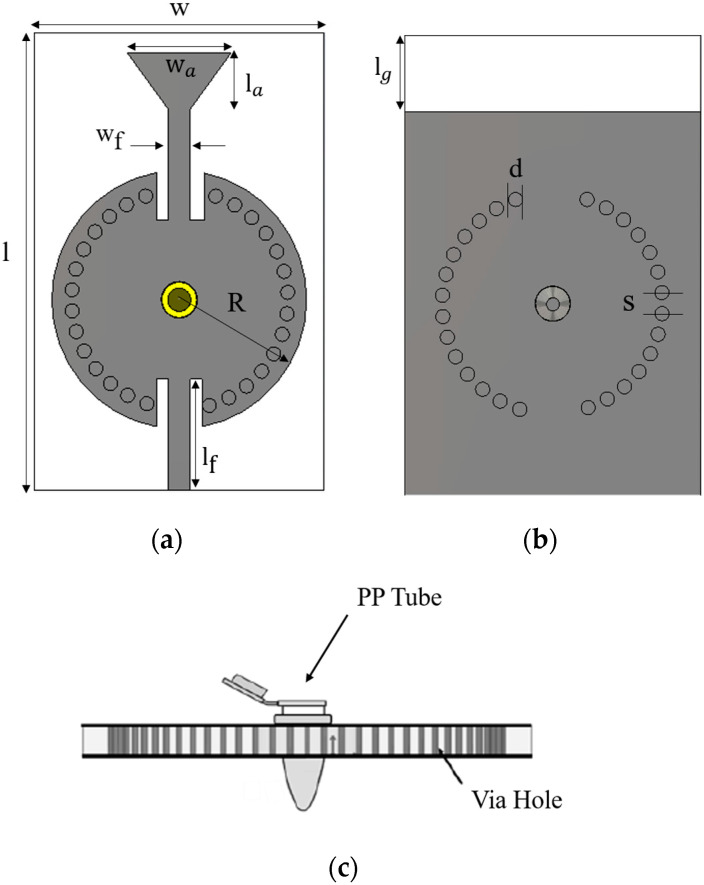
Designed layout of the proposed sensor: (**a**) front view, (**b**) back view, and (**c**) side view.

**Figure 2 sensors-23-07835-f002:**
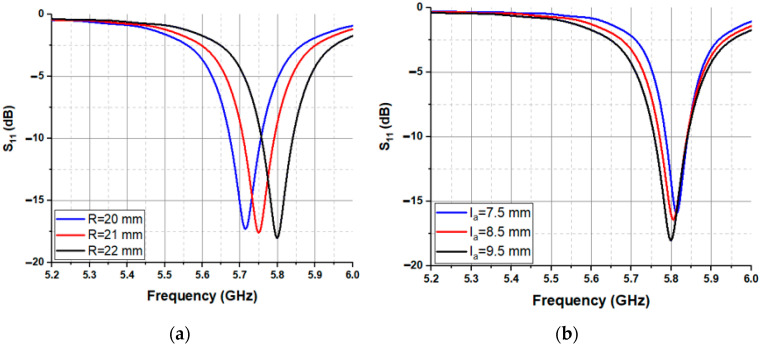
Parametric study of the antenna: (**a**) radius of the circular SIW (R), (**b**) length of the antenna ($l_{a}$).

**Figure 3 sensors-23-07835-f003:**
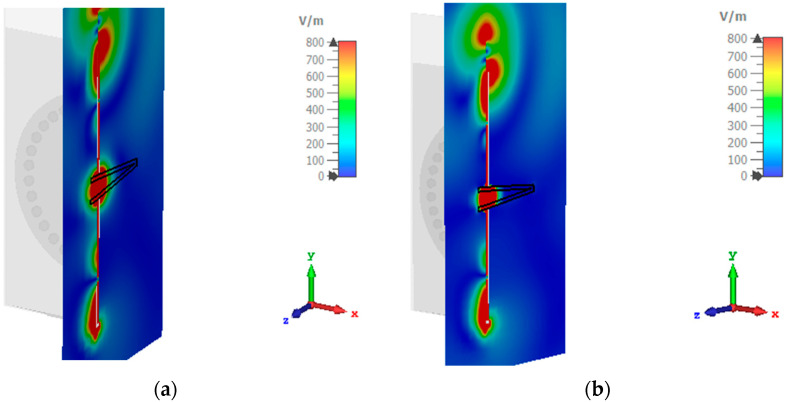
Electric field distribution of the proposed structure: (**a**) without metal sheath, (**b**) with metal sheath.

**Figure 4 sensors-23-07835-f004:**
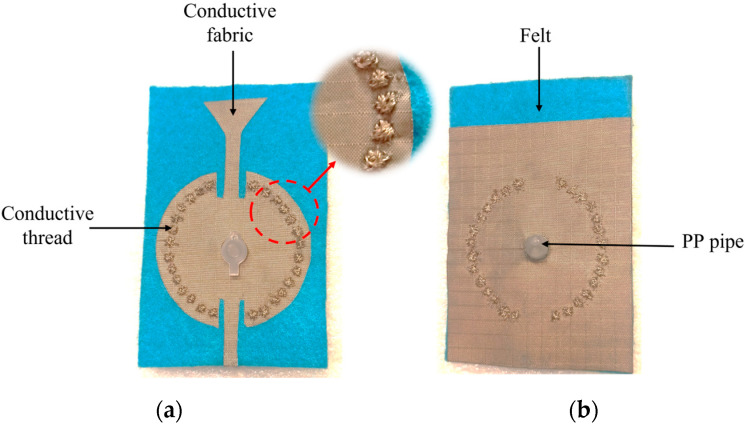
Photograph of the manufactured proposed system: (**a**) front view, (**b**) back view.

**Figure 5 sensors-23-07835-f005:**
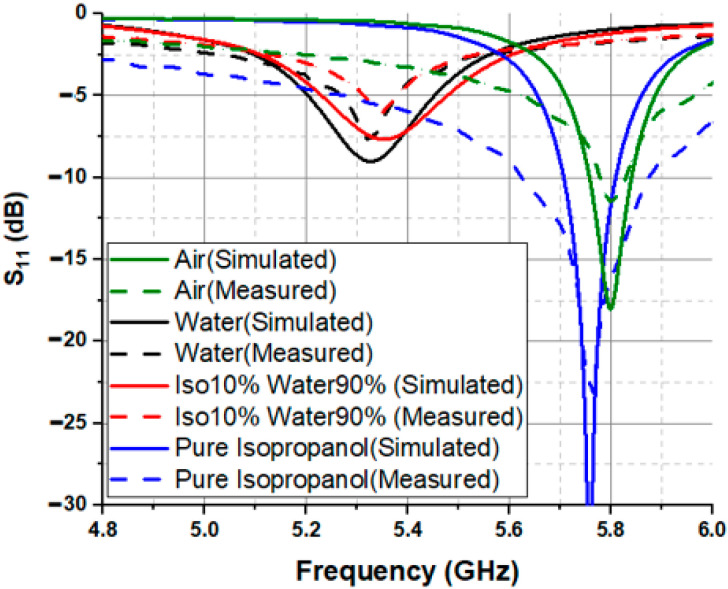
Simulated and measured S_11_ for air, distilled water, pure isopropanol, and a mixture of 90% isopropanol and 10% water.

**Figure 6 sensors-23-07835-f006:**
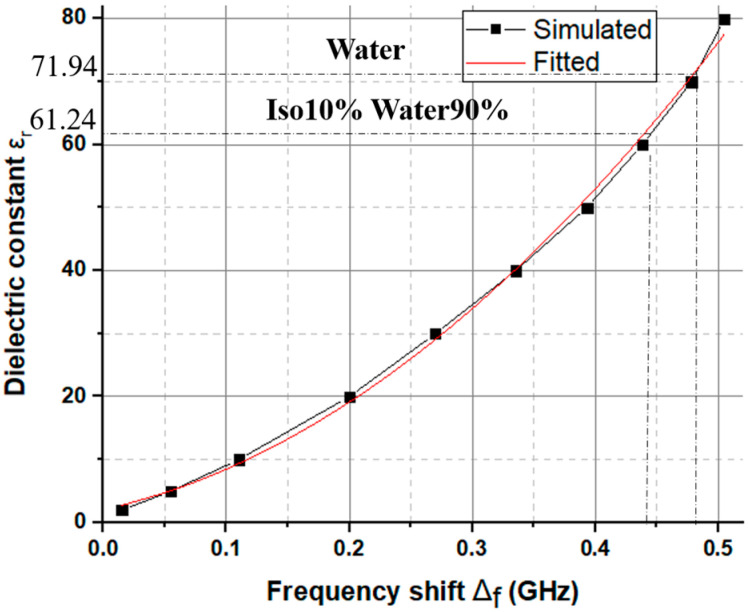
Simulated values of the dielectric constant versus the resonance frequency shift, and experimental validation examples.

**Figure 7 sensors-23-07835-f007:**
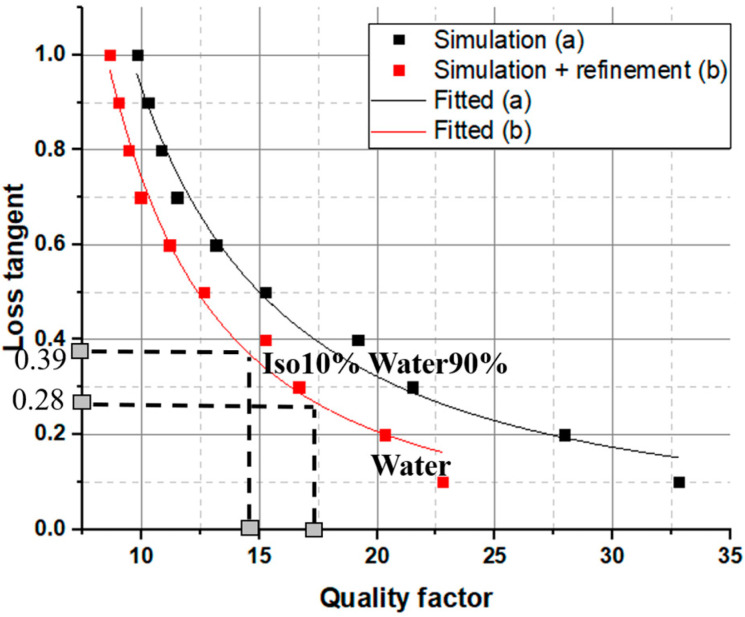
Simulated values of the loss tangent versus the unloaded quality factor for the experimental validation examples using water and a mixture of 10% Iso 90% water, (a) Simulation, (b) Simulation + refinement.

**Table 1 sensors-23-07835-t001:** Detailed geometric parameters of the proposed structure.

Parameter	Dimension (mm)	Description
W	50	Width of substrate and bottom ground
L	78	Length of substrate and bottom ground
$w_{a}$	17.84	Width of the antenna
$l_{a}$	9.5	Length of the antenna
R	22	Radius of the circular SIW
$l_{f}$	18.98	Length of feeding line
$w_{f}$	3.84	Width of feeding line
$l_{g}$	13	Length of the partial ground
D	2.4	Diameter of SIW vias
S	3.5	Pitch distance between vias

**Table 2 sensors-23-07835-t002:** Dielectric constant retrieved with the proposed technique and reference values measured with the coaxial probe.

Mixture under Test	SIW Antenna Sensor	Coaxial Probe	Relative Error %
Pure water	71.94	72.44	0.69
10% Iso 90% Water	61.24	61.37	0.21
30% Iso 70% Water	38.84	37.92	2.42
50% Iso 50% Water	23.72	22.52	5.32
70% Iso 30% Water	14.23	12.10	17.60
90% Iso 10% Water	6.37	5.70	11.75
Pure Isopropanol	4.36	3.89	12.08

**Table 3 sensors-23-07835-t003:** Loss tangent retrieved with the proposed technique and reference values.

Mixture under Test	SIW Antenna Sensor	Coaxial Probe	Relative Error %
Pure water	0.28	0.29	3.44
10% Iso 90% Water	0.39	0.41	4.87
30% Iso 70% Water	0.69	0.67	2.98
50% Iso 50% Water	0.84	0.79	6.32
70% Iso 30% Water	0.82	0.81	1.23
90% Iso 10% Water	0.72	0.64	12.50
Pure Isopropanol	0.37	0.35	5.71

**Table 4 sensors-23-07835-t004:** Comparison of the proposed sensor with other published work for liquid characterization.

Ref.	Sensor Type	$f_{air}$ (GHz)	$\varepsilon_{r}$ Range	Sensitivity (GHz/Δ$\varepsilon_{r}$)	Fabrication Technology
[21]	Complementary split ring resonator (CSRR)	2.37	9–79	0.07	PCB
[6]	Sensor based on SIW cavity	3.82	4–76	0.57	3D printed
[22]	Microstrip-coupled CSRR	2	9–79	0.6	PCB
[23]	SIW cavity resonator	17.08	2–79	0.04	PCB
[24]	SIW rectangular slot antenna sensor	8.96	8–66	0.65	PCB
[9]	Microwave SIW cavity	2.45	25–90	0.46	PCB
This work	SIW circular cavity-based antenna sensor	5.8	4–72	0.7	Textile

## Data Availability

Not applicable.

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
