# Peer review of "Textile Antenna Sensor in SIW Technology for Liquid Characterization"

_sensors, 2023, doi:10.3390/s23187835_

Round 1

Reviewer 1 Report

This work presents the development of a textile-based SIW sensor for characterization of dielectric properties of two liquids: IPA and water, and their mixtures.  I have a few questions and comments as follows which may help make the overall manuscript more easily understood by other readers.

- "The geometry of the circular SIW structure is designed by ensuring 94 optimal space between vias to minimize the leakage loss of SIW" How was this optimal space determined?  What optimization process was done?

- How accurately were the design dimensions transferred to the fabric and what was the process used?

- I believe Figure 2 would be best represented with either (or both) a cross-sectional view and a top-down view of the sheath/no-sheath field distribution.  The current diagonal view uses different angles and is hard to compare.

- There are still large differences between simulated and measured S11 responses (Fig.4) in quality factor and resonant amplitude, couldn't the loss tangent estimates be used to make these curves fit more closely since that is similar to how they are calculated initially?

- I'm curious how the flexibility of the substrate  contributed to potential errors or uncertainties in the calculations due to potential movement between measurements  ~ Was this investigated at all?

- Another potential work which might be worth discussing as resonator-textile integration: "Oleophobic textiles with embedded liquid and vapor hazard detection using differential planar microwave resonators"

English is good, very few mistakes ("yarnis", page 3)

Author Response

First of all, we appreciate the reviewers time and effort to provide their valuable feedback on the article. We are grateful for their insightful comments on the article. We have incorporated the changes to reflect the suggestions provided by the reviewers. We have highlighted the changes within the manuscript.

Reviewer #1:

We appreciate the comments from the reviewer and our responses are shown as follows:

This work presents the development of a textile-based SIW sensor for characterization of dielectric properties of two liquids: IPA and water, and their mixtures.  I have a few questions and comments as follows which may help make the overall manuscript more easily understood by other readers.

  1. "The geometry of the circular SIW structure is designed by ensuring 94 optimal spaces between vias to minimize the leakage loss of SIW"How was this optimal space determined?  What optimization process was done?

Response 1:

We appreciate the reviewer’s comment. The proposed structure was designed by choosing appropriately spaced via holes with the same diameter. There are two design rules related to via diameter ''D'' and spacing between the vias "P" called pitch as given by [4]:

The optimal space was determined using these rules to minimize the leakage loss of SIW structure. The geometry and dimensions of the SIW structure are based on simulation results and parameter sweeps. Use optimization techniques by CST to find the best set of parameters that meet the design specifications.

This paragraph has been included in Section 2 (Line 97-101).

  1. How accurately were the design dimensions transferred to the fabric and what was the process used?

Response 2:

The accuracy of transferring design dimensions to fabric depends on the specific process and technique used in the manufacturing. The embroidery technique was used to manufacture the vias of the SIW structure. This method consists to export the simulated structure from CST software to a file format readable by the design embroidery software (Easy Design EX4.0). The computerized embroidery process was implemented by means of a satin fill pattern creating vias of stitches. The stitch used a needle yarn passing via the substrate and overlapping with a bobbin yarn. The computerized embroidery machine ensures precise design replication of our SIW structure. For antenna SIW sensor manufacturing, the embroidery machine Singer Futura XL 550 was used.

This paragraph has been included in Section 2 (Line 113-120).

  1. I believe Figure 2 would be best represented with either (or both) a cross-sectional view and a top-down view of the sheath/no-sheath field distribution.  The current diagonal view uses different angles and is hard to compare.

Response 3:

We appreciate the reviewer’s comment. The figure 2 presents the electric field distribution of the proposed structure. It can be clearly seen that after adding the metal sheath the electromagnetic field penetrated more deeply inside the PP pipe.

  1. There are still large differences between simulated and measured S11 responses (Fig.4) in quality factor and resonant amplitude, couldn't the loss tangent estimates be used to make these curves fit more closely since that is similar to how they are calculated initially?

Response 4:

Figure 4 shows the calculated and measured S11 responses, this result shows the initial S11 response from four different cases, to give readers an idea before retrieve the dielectric properties.

  1. I'm curious how the flexibility of the substrate contributed to potential errors or uncertainties in the calculations due to potential movement between measurements ~ Was this investigated at all?

Response 5:

We appreciate the reviewer’s comment. We recognized the importance of substrate movement between measurements and took several measures to address this potential source of error, such as securing the substrate with specialized supports and clamps to minimize any unintended movement during measurements.

  1. Another potential work which might be worth discussing as resonator-textile integration: "Oleophobic textiles with embedded liquid and vapor hazard detection using differential planar microwave resonators".

Response 6:

In [16], a textile microwave resonator sounds intriguing and highly relevant. The integration of resonator technology into textile presents a responsive system capable of sensing to liquid. This could have a significant impact on various application, including environmental monitoring, safety and healthcare.

This paragraph has been included in Section 1 (Line 56-59).

Reviewer 2 Report

This paper has a great textual and some technical similarity (more than 8%) with the " G. M. Rocco et al., "3-D Printed Microfluidic Sensor in SIW Technology for Liquids’ Characterization," in IEEE Transactions on Microwave Theory and Techniques, vol. 68, no. 3, pp. 1175-1184, March 2020, doi: 10.1109/TMTT.2019.2953580." paper, which is 45% in total. Please reconsider the presentation and organization of the paper. This can lead to an academic ethical problem. After these rates are lowered, the paper can be evaluated technically.

Dear Editor

This paper has a great textual and some technical similarity (more than 8%) with the " G. M. Rocco et al., "3-D Printed Microfluidic Sensor in SIW Technology for Liquids’ Characterization," in IEEE Transactions on Microwave Theory and Techniques, vol. 68, no. 3, pp. 1175-1184, March 2020, doi: 10.1109/TMTT.2019.2953580." paper, which is 45% in total. Please reconsider the presentation and organization of the paper. This can lead to an academic ethical problem. After these rates are lowered, the paper can be evaluated technically.

BR.

Assoc Prof. MHBilgehan UCAR

Author Response

First of all, we appreciate the reviewers time and effort to provide their valuable feedback on the article. We are grateful for their insightful comments on the article. We have incorporated the changes to reflect the suggestions provided by the reviewers. We have highlighted the changes within the manuscript.

Reviewer #2:

This paper has a great textual and some technical similarity (more than 8%) with the " G. M. Rocco et al., "3-D Printed Microfluidic Sensor in SIW Technology for Liquids’ Characterization," in IEEE Transactions on Microwave Theory and Techniques, vol. 68, no. 3, pp. 1175-1184, March 2020, doi: 10.1109/TMTT.2019.2953580." paper, which is 45% in total. Please reconsider the presentation and organization of the paper. This can lead to an academic ethical problem. After these rates are lowered, the paper can be evaluated technically.

Response:

The novelty of the proposed system compared to the reference '" G. M. Rocco et al., "3-D Printed Microfluidic Sensor in SIW Technology for Liquids’ Characterization," that our sensor is a fully textile embroidered antenna sensor for liquid characterization. To the best of our knowledge, this paper demonstrates for the first time the capability of a fully-textile antenna sensor to characterize liquids. In addition, this paper is a collaborative work with the same lab. So, there is no academic ethical problem. The main idea is to use the method presented in the cited reference to retrieve the dielectric properties of liquid under test using a fully textile system.

Reviewer 3 Report

The authors present a novel textile antenna sensor using a resonant cavity for liquid characterization. With a hole in the middle and a pipe passing through, the cavity uses circular substrate integrated waveguide technology (SIW).

1- From Figure 3 I can't know How you can be made the vias of SIW from the Conductive thread. 

2- Are the pipe material taken into consideration?

3- Why the authors did not use the difference in phase in calculations and depended on the delta F only

4- What is about radiation pattern of the proposed antenna.

5- The authors mentioned the coaxial probe characterization technique. Are you mean the Dielectric Assistant Kit?

6- Please, explain why the error in  90% Iso 10% Water case is 12% while in 10% Iso 90% Water case is 4.87.

7- The references are not updated. There is just one reference in 2023.

Thanks

Author Response

First of all, we appreciate the reviewers time and effort to provide their valuable feedback on the article. We are grateful for their insightful comments on the article. We have incorporated the changes to reflect the suggestions provided by the reviewers. We have highlighted the changes within the manuscript.

Reviewer #3:

We appreciate the comments from the reviewer and our responses are shown as follows:

The authors present a novel textile antenna sensor using a resonant cavity for liquid characterization. With a hole in the middle and a pipe passing through, the cavity uses circular substrate integrated waveguide technology (SIW).

  1. From Figure 3 I can't know How you can be made the vias of SIW from the Conductive thread.

Response 1:

We appreciate the reviewer’s comment. The vias for the SIW using conductive thread is a multi-step process using the embroidery machine. During the embroidery, the conductive yarn is carefully positioned on the substrate to define the shape and path of the desired vias. This can be achieved by using precision placement tools of the embroidery machine to ensure precise alignment.

This paragraph has been included in Section 2 (Line 108-110).

  1. Are the pipe material taken into consideration?

Response 2:

Note that the pipe material is taken into consideration when designing and implementing the system to ensure they meet the exact requirement used in the measurement.

This paragraph has been included in Section 2 (Line 143-145).

  1. Why the authors did not use the difference in phase in calculations and depended on the delta F only

Response 3:

We appreciate the reviewer’s comment. Different approaches are used in the literature to retrieve the liquid characterization such as approach based on the variation of the transmission amplitude. In this paper, the approach used is based on the shift of the resonance frequency to retrieve the dielectric properties (dielectric constant and tangent loss) of the material under test.

  1. What is about radiation pattern of the proposed antenna.

Response 4:

Note that the antenna used in the system is a broadband antenna that can be used to transmit data for the retrieved dielectric properties of liquid under test. This part will be included in future work.

This paragraph has been included in Section 4 (Line 244-247).

  1. The authors mentioned the coaxial probe characterization technique. Are you mean the Dielectric Assistant Kit?

Response 5:

For comparative purposes, the dielectric properties of different liquids are measured using a commercial open-ended coaxial probe Keysight N1501A Dielectric Probe Kit, which is used as the reference technique.

This paragraph has been included in Section 3 (Line 189-191).

  1. Please, explain why the error in 90% Iso 10% Water case is 12% while in 10% Iso 90% Water case is 4.87.

Response 6:

The samples of 90% Iso 10% Water and 10% Iso 90% Water might inherently vary in their composition or other characteristics, which could impact the accuracy of measurements or calculations.

  1. The references are not updated. There is just one reference in 2023.

We appreciate the reviewer’s comment. The references are updated by adding new references. Ref [8] and [15] .

Round 2

Reviewer 2 Report

·         I am aware that your work is different from the paper I mentioned (G. M. Rocco et al., "3-D Printed Microfluidic Sensor in SIW Technology for Liquids’ Characterization," in IEEE Transactions on Microwave Theory and Techniques, vol. 68, no. 3, pp. 1175-1184, March 2020, doi: 10.1109/TMTT.2019.2953580). However, there are textual similarities with this paper and many publications in the literature. Please reduce these rates according to the attached turnitin report.

·         The last paragraph of the Introduction section should mention the organization of the rest of the article. Please add a paragraph about the organization of the paper.

·         Why was the antenna developed for liquid characterization designed as textil-based? How is it different from other pcb based designs? The reason for this should be emphasized both in the abstract and in the introduction sections.

·         Which fabrication method did you use? What is the sensitivity of the fabrication method you use? Have you taken this sensitivity into account in the differences between measurement and simulation results?

·         It is stated that the pipe where you put the liquid is made of polypropylene material. What was the electrical permeability of the polypropylene material in the simulations? In realistic implementation, could you confirm the electrical properties of this material before the measurements?

·         In the proposed sensor design, how did you determine the dimensions R, d, S wa and la ? Have you used/followed any optimization process?

·         Parameteric results related to critical design parameters regarding the effect of physical parameters in sensor design should be included in the paper.

Author Response

First of all, we appreciate the reviewer's time and effort to provide his valuable feedback on the article. We are grateful for his insightful comments on the article. We have incorporated the changes to reflect the suggestions provided by the reviewer. We have highlighted the changes within the manuscript.

Reviewer #2:

We appreciate the comments from the reviewer and our responses are shown as follows:

  1. I am aware that your work is different from the paper I mentioned (G. M. Rocco et al., "3-D Printed Microfluidic Sensor in SIW Technology for Liquids’ Characterization," in IEEE Transactions on Microwave Theory and Techniques, vol. 68, no. 3, pp. 1175-1184, March 2020, doi: 10.1109/TMTT.2019.2953580). However, there are textual similarities with this paper and many publications in the literature. Please reduce these rates according to the attached turnitin report.

Response 1:

The changes are highlighted in green in the manuscript.

  1. The last paragraph of the Introduction section should mention the organization of the rest of the article. Please add a paragraph about the organization of the paper.

Response 2:

The remainder of the paper is organized as follows. Section 2 details the methodology of the proposed design, the sensing principle, and the materials used in the manufacturing. Section 3 provides the results of the proposed SIW antenna sensor. A comparison of the proposed SIW antenna sensor with previously reported works in the literature for liquid characterization is presented in section 4. Finally, Section 5 exhibits the main conclusions.

This paragraph has been included in Section 1 (Line 101-106).

  1. Why was the antenna developed for liquid characterization designed as textil-based? How is it different from other pcb based designs? The reason for this should be emphasized both in the abstract and in the introduction sections.

Response 3:

We appreciate the reviewer’s comment, To the best of our knowledge, this article demonstrates for the first time the ability of a fully textile SIW cavity antenna-based sensor to characterize the dielectric properties of a liquid under test and emphasizes its differentiating features compared to PCB-based designs. The unique attributes of the textile-based antenna stem from its flexibility, conformability, and compatibility with various liquids.

This paragraph has been included in Abstract (Line 23-27).

The development of textile-based antennas for liquid characterization is entrenched in their exceptional flexibility and conformability. Unlike traditional PCB-based antennas, which are inherently rigid and often ill-suited for conforming to irregular shapes or directly interfacing with liquids, textile-based antennas offer remarkable pliability. This enables seamless integration onto various substrates, including curved surfaces and complex geometries, facilitating direct and efficient interaction with the liquids under scrutiny.

This paragraph has been included in Section 1 (Line 58-64).

  1. Which fabrication method did you use? What is the sensitivity of the fabrication method you use? Have you taken this sensitivity into account in the differences between measurement and simulation results?

Response 4:

We appreciate the reviewer’s comment, the embroidery technique was used to manufacture the vias of the SIW structure. This method consists to export the simulated structure from CST software to a file format readable by the design embroidery software (Easy Design EX4.0). The computerized embroidery process was implemented by means of a satin fill pattern creating vias of stitches. The stitch used a needle yarn passing via the substrate and overlapping with a bobbin yarn.  The computerized embroidery machine ensures precise design replication of our SIW structure. For antenna SIW sensor manufacturing, the embroidery machine Singer Futura XL 550 was used.

This paragraph has been included in Section 2 (Line 126-133).

The sensitivity of the fabrication method is +/- 0.5mm. The sensitivity analysis is a crucial tool for understanding how variations in input parameters affect the outcomes of simulations and measurements. It helps to identify potential sources of discrepancies and aids in refining models to better match real-world data.

  1. It is stated that the pipe where you put the liquid is made of polypropylene material. What was the electrical permeability of the polypropylene material in the simulations? In realistic implementation, could you confirm the electrical properties of this material before the measurements?

Response 5:

We appreciate the reviewer’s comment, the electrical properties of the polypropylene material used in the simulation are:

Dielectric Constant: .

Loss tangent: .

The values of the electrical properties are taken from the data sheet of the pipe. During the measurements, the results are quite similar to the simulation which means the electrical properties used in the simulation match with the realistic implementation.

  1. In the proposed sensor design, how did you determine the dimensions R, d, S wa and la? Have you used/followed any optimization process?

Response 6:

The geometry and dimensions of the SIW structure are based on simulation results and parameter sweeps. Use optimization techniques by CST to find the best set of parameters that meet the design specifications.

  1. Parametric results related to critical design parameters regarding the effect of physical parameters in sensor design should be included in the paper.

Response 7:

A parametric study is accomplished using a geometric tuning approach on different parameters of the proposed antenna sensor in parameters in order to obtain better insight into the antenna's physical performance and to fine-tune its behavior specifically for the 5.8 GHz frequency. The S11 for a variation of radius of the circular SIW (R) and length of the antenna () are shown in Fig.2.

This paragraph has been included in Section 2 (Line 147-151).

Figure 2. Parametric study of the antenna: (a) radius of the circular SIW (R), (b) length of the antenna ().
